# Interfacial ice sprouting during salty water droplet freezing

Fuqiang Chu [1,2], Shuxin Li[1], Canjun Zhao [3], Yanhui Feng [1,2] ✉, Yukai Lin[3], Xiaomin Wu [3] ✉, Xiao Yan[4,5,6] ✉ & Nenad Miljkovic [6,7,8,9] ✉

Icing of seawater droplets is capable of causing catastrophic damage to vessels, buildings, and human life, yet it also holds great potential for enhancing applications such as droplet-based freeze desalination and anti-icing of sea sprays. While large-scale sea ice growth has been investigated for decades, the icing features of small salty droplets remain poorly understood. Here, we demonstrate that salty droplet icing is governed by salt rejection-accompanied ice crystal growth, resulting in freezing dynamics different from pure water. Aided by the observation of brine films emerging on top of frozen salty droplets, we propose a universal definition of freezing duration to quantify the icing rate of droplets having varying salt concentrations. Furthermore, we show that the morphology of frozen salty droplets is governed by ice crystals that sprout from the bottom of the brine film. These crystals grow until they pierce the free interface, which we term ice sprouting. We reveal that ice sprouting is controlled by condensation at the brine film free interface, a mechanism validated through molecular dynamics simulations. Our findings shed light on the distinct physics that govern salty droplet icing, knowledge that is essential for the development of related technologies.

Droplet icing is a ubiquitous and fascinating phenomenon found in nature and our daily lives. When a water droplet rests on a cold surface whose temperature is below the freezing point, it solidifies into an ice pellet. In industry, icing is usually not desired because it can cause damage to several processes such as crop production, power generation, building infrastructure, refrigeration, aviation, and navigation[1–5]. For pure water droplet icing, the nucleation, recalescence, and freezing stages take place in sequence[6–8]. Nucleation refers to ice nucleus formation, whether it be homogeneous or heterogeneous, and recalescence represents rapid ice crystal growth and temperature rise[9–11]. Freezing is manifested as ice front propagation, which is mainly controlled by thermal transport[12–14]. In recent years, the understanding of water droplet icing has expanded significantly, with a variety of interesting phenomena discovered, such as tip formation[15], droplet explosion[16], pattern generation[17], bubble formation[18], and re-icing[19], to name a few. As a result, various anti-icing methods have been proposed using nucleation delay, adhesion reduction, and ice self-removal[20–30].

The icing of pure water droplets is complex. Adding additives to pure water causes additional unexpected icing phenomena[31]. For instance, frozen nanofluid droplets exhibit flat plateaus on top instead of pointy tips[32]. Adding a soluble salt, such as sodium chloride (NaCl),

[1]School of Energy and Environmental Engineering, University of Science and Technology Beijing, Beijing 100083, China. [2]Beijing Key Laboratory of Energy Conservation and Emission Reduction for Metallurgical Industry, University of Science and Technology Beijing, Beijing 100083, China. [3]Department of Energy and Power Engineering, Tsinghua University, Beijing 100084, China. [4]Key Laboratory of Low-grade Energy Utilization Technologies and Systems, Chongqing University, Ministry of Education, Chongqing 400030, China. [5]Institute of Engineering Thermophysics, Chongqing University, Chongqing 400030, China. [6]Department of Mechanical Science and Engineering, University of Illinois at Urbana–Champaign, Urbana, IL 61801, USA. [7]Department of Electrical and Computer Engineering, University of Illinois at Urbana–Champaign, Urbana, IL 61801, USA. [8]Materials Research Laboratory, University of Illinois at Urbana–Champaign, Urbana, IL 61801, USA. [9]International Institute for Carbon Neutral Energy Research (WPI-I2CNER), Kyushu University, 744 Moto-oka, Nishi-ku, Fukuoka 819-0395, Japan. ✉e-mail: yhfeng@me.ustb.edu.cn; wuxiaomin@mail.tsinghua.edu.cn; yan23@cqu.edu.cn; nmiljkov@illinois.edu

to the droplet makes the tip disappear[33]. The addition of salt also enables anti-icing by reducing the ice nucleation temperature due to the increased ice-fluid interface formation energy barrier, which may be related to the restructuring of bound water in the formation of hydration shells of ions[34], and slowing the freezing process due to brine repulsion[35,36]. At extremely low temperatures, NaCl dihydrate forms in microdroplets, and the effloresced NaCl can act as ice nucleating particles[37,38]. Although some advances in the understanding of nucleation kinetics and ion repulsion have been reported, salty water droplet icing remains poorly understood. What are the unique features of salty droplet icing, how does the freezing morphology of salty droplets differ from that of pure water droplets, and how can one determine and predict the freezing time of salty droplets? The answers to these questions, as well as the revelation of the underlying mechanisms, are knowledge gaps that need to be filled.

In this work, we provide comprehensive insights into salty droplet icing. We demonstrate a significant difference between icing of salty water droplets and pure water droplets through the combination of experiments, theory, and molecular dynamics (MD) simulations. We define the freezing time of salty droplets based on both temperature measurement and optical imaging. We show that at the end of the salty droplet freezing, a liquid film forms on top of the frozen droplet, inside of which ice sprouts, grows, and eventually punctures the film. We not only reveal the mechanisms governing this interesting ice sprouting phenomenon, which is highly dependent on condensation, but also reproduce the phenomenon using MD simulations to verify the mechanism. The unique insights provided here elucidate the rich physics governing ice crystal growth and brine expulsion, as well as the complex interactions between icing and condensation phase change. Our findings enhance the understanding of both natural and industrial icing of salty water, such as freeze desalination[39] and marine icing[40].

## Results
### Icing characteristics of salty droplets
We conduct icing experiments of salty water droplets residing on solid surfaces. The droplets are mainly made from NaCl aqueous solutions having concentrations (mass fraction, $\omega$) ranging from 0 to 16%. The volume of deposited salty droplets ($V$) ranges from 4 to 40 μL. For better visualization of the droplets, hydrophobic surfaces are used as the substrates, and the surfaces are cooled to a temperature ($T_w$) ranging from −10 to −30 °C by a refrigeration device. In most cases, the surface temperature $T_w$ is higher than the eutectic point temperature of NaCl solution (−21.1 °C). The icing experiments are performed in an open environment whose air temperature ($T_{air}$) is 25.0 ± 3 °C, and relative humidity ($RH$) ranges from 30 to 95%. All of the experimental conditions are representative of conventional natural icing conditions that are usually adopted in past literature[6,26]. Extreme conditions, such as experiments with high cooling rates are not considered. See Supplementary Information S1 and S2 for more details regarding surface preparation and the experimental setup.

We first characterize the icing process of a pure water droplet (deionized water, $V = 4$ μL). Figure 1A shows the nucleation/recalescence stage followed by the freezing stage during icing of a pure water droplet. The freezing stage is characterized by orderly ice front propagation and ends with a pointy tip at the top of the frozen droplet. The two stages are also obviously distinguished in the temperature curve during icing (Supplementary Information S3, Figure S3a). Conducting the same experiment using a salty water droplet ($\omega = 9\%$, $V = 4$ μL) reveals that the same two-step process is followed, as confirmed using temperature mapping of the droplet (Figure S3a). However, the nucleation/recalescence stage lasts one order of magnitude longer than that observed on an equivalent size pure water droplet. The freezing stage is also delayed. Furthermore, the observed time scale difference is reflected in the visualization of the icing process, but the boundary between the two stages is no longer obvious for salty droplet icing (Fig. 1B). This is due to

both nucleation/recalescence and freezing being manifested by the growth of ice crystals inside of the salty droplet. During the nucleation/recalescence stage, the ice crystals grow faster due to larger supercooling. During the freezing stage, the ice crystals grow slowly and haphazardly so that one cannot observe regular ice front motion inside the three-dimensional salty droplet. The perspective view in Fig. 1C clearly shows the icing characteristics of salty droplets.

Since the ice crystal growth is accompanied by brine repulsion and concentration[35], the salty droplets will not completely freeze (i.e., concentrated brines exist between ice dendrites, as depicted in Fig. 1C). This observation is supported by our MD simulation results. See Supplementary Information S6 for MD simulation details. As the ice nucleates and grows, most of the sodium and chloride ions are expelled from the ice, with only a few trapped within the ice (Fig. 1D), resulting in an increasingly concentrated remaining liquid. After some time (300 ns), the number of frozen water molecules no longer increases significantly, and the salt concentration of the unfrozen solution does not change (Fig. 1E, F). To verify these mechanisms, we conducted macroscopic experiments to observe the ice crystals and unfrozen brines inside the droplet. Figure 1G shows images of a frozen salty droplet within a Hele-Shaw cell. The results show that concentrated brines exist in the crevices of ice dendrites, i.e., the frozen droplet is a mixture of ice and concentrated brine. The basic phase-change characteristics (e.g., ice crystal growth and brine expulsion) during salty droplet freezing on a solid surface are like that of sea ice growth, except that sea ice grows from the sea surface downwards, leaving brine channels inside the ice[41,42]. Salty droplets heterogeneously nucleate from the solid surface, and ice crystals grow upward towards the liquid-vapor interface.

### Freezing time prediction of salty droplets
Predicting the duration of salty droplet freezing is of practical importance to engineering applications. Since the salty droplet is not completely frozen, we define a freezing duration time ($t_{fr}$) for salty droplets. We regard the moment of nucleation as zero time using the temperature response measured by the thermocouples inserted inside the salty droplet (Fig. 2A). After the transitory nucleation/recalescence stage, the droplet temperature decreases sharply until an inflection point is reached, after which the droplet temperature stabilizes. Taking the derivative of the temperature curve, we obtained the inflection point where the change rate of temperature, i.e., the derivative of the temperature curve, approaches zero (Fig. 2B). See Supplementary Information S4 for a detailed determination of the inflection points. For pure water droplet icing, it is well known that the freezing stage ends with the appearance of a pointy tip on top of the droplet[15]. Also, an inflection point is established in the temperature when the tip is formed (Figure S3a). Therefore, we define the freezing time of a salty droplet, $t_{fr}$, as the duration from nucleation time to the inflection point in the temperature curve (since the nucleation/recalescence stage is much shorter than the freezing stage, we initiate the freezing time from the moment of nucleation). Under the same conditions, we demonstrate that the freezing time obtained from multiple temperature curves is consistent (Fig. 2A, B, and Figure S4). This not only indicates the universality of the defined freezing time by the inflection point, but also confirms that the inflection point and measured freezing time are independent of the position of the inserted thermocouples (Supplementary Information S5, Figure S5).

Based on established knowledge, we expected the absence of the pointy tip on the salty droplet. However, we were surprised to observe the formation of a liquid film on top of the droplet after the freezing period. As shown in Fig. 2C, the reflection of the light source at the top of the droplet gradually becomes blurred before 34 s. After 34 s, the blurry light source reflection becomes clear again, indicating that a liquid film is formed at the top of the droplet. Supplementary Movie 2 shows the brine film formation process more clearly. By comparing the

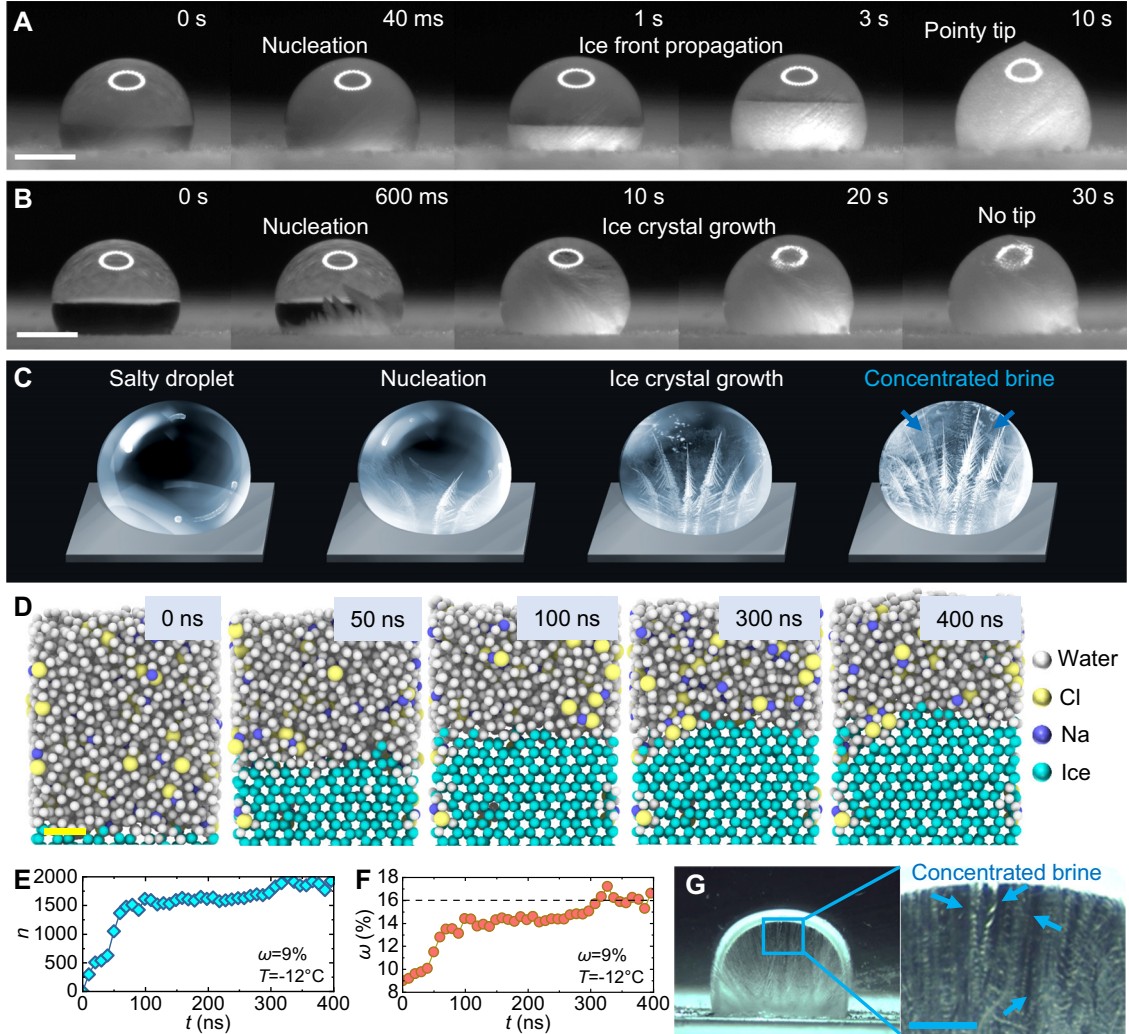

**Fig. 1 | Characteristics of salty droplet icing. A** Icing of a pure water droplet ($V = 4$ μL and $T_w = -15$ °C). The nucleation/recalescence stage and the freezing stage are clearly shown. **B** Icing of a salty droplet ($V = 4$ μL, $\omega = 9\%$, and $T_w = -15$ °C). See Supplementary Movie 1. The boundary between the two stages is not easily recognized. Scale bars for A and B represent 1 mm. **C** Schematic of salty droplet icing. Ice crystals grow during the entire icing process, and the brine is repelled and concentrated gradually. **D** Time-lapse snapshots of a salty water icing MD simulation. The system temperature is set to 261 K ($T = -12$ °C), and the original concentration of salty water is $\omega = 9\%$. The majority of sodium and chloride ions are expelled from the ice front. Scale bar in **D** represents 1 nm. **E** Number of icing water molecules ($n$) and **F** salt concentration ($\omega$) in the remaining unfrozen liquid as a function of time ($t$) extracted from the MD simulation. **G** Salty droplet freezing as observed in a Hele-Shaw cell. The two-dimensional images demonstrate that a concentrated brine exists in the gaps between ice dendrites. Scale bar in **G** represents 1 mm. Source data are provided as a Source Data file.

experimental images with the temperature curve, we find that the liquid film on the top of the droplet begins to appear (see the image at 34 s in Fig. 2C or inset at 35 s in Fig. 2A) immediately at the temperature curve inflection time (red circle in Fig. 2A). The moment when the liquid film appears coincides with the end of the freezing stage. Therefore, one can determine the freezing time of a salty droplet not only from its temperature curve but also from visualization.

The freezing time is measured under varying conditions including $V$, $\omega$, and $T_w$ (Fig. 2D, E). As $V$, $\omega$, and $T_w$ increase, $t_{fr}$ increases. Although disorganized ice crystal growth occurs instead of orderly ice front propagation during freezing, we observe from two-dimensional images within the Hele-Shaw cell that the ice crystals grow densely but independently (Supplementary Information S7, Figure S8). For each ice crystal aggregate within a salty droplet (Figure S9), we can approximate its growth by the Stefan problem[18,43]. Under a typical condition with supercooling of $\Delta T \sim 15$ °C, the Stefan number, $St = C_{p,ice}\Delta T/L_{mix} = \tau_d/\tau_{sol}$ is <0.1 (here $C_{p,ice}$ is the specific heat capacity of ice and $L_{mix}$ is the latent heat of solidification; $\tau_d$ is the time scale of

thermal diffusion in ice and $\tau_{sol}$ is the time scale of solidification). The low $St$ indicates that $\tau_d$ is much smaller than $\tau_{sol}$. See more detailed calculations in Supplementary Information S8. Therefore, the heat transfer in an ice crystal aggregate can be approximated using one-dimensional quasi-steady heat transfer:

$$\frac{k_{ice}\Delta T}{H} = \rho_{ice}L_{mix}\frac{dH}{dt}, \quad (1)$$

where $k_{ice}$ is the ice thermal conductivity, $\rho_{ice}$ is the ice density, and $H$ is the freezing front height defined by the length of ice crystal aggregate. Here, the supercooling $\Delta T$ is the difference between the freezing temperature at the ice crystal front and the surface temperature. Integration of Eq. (1) yields an expression for $t_{fr}$:

$$t_{fr} = \frac{\rho_{ice}L_{mix}H_f^2}{2k_{ice}\Delta T}, \quad (2)$$

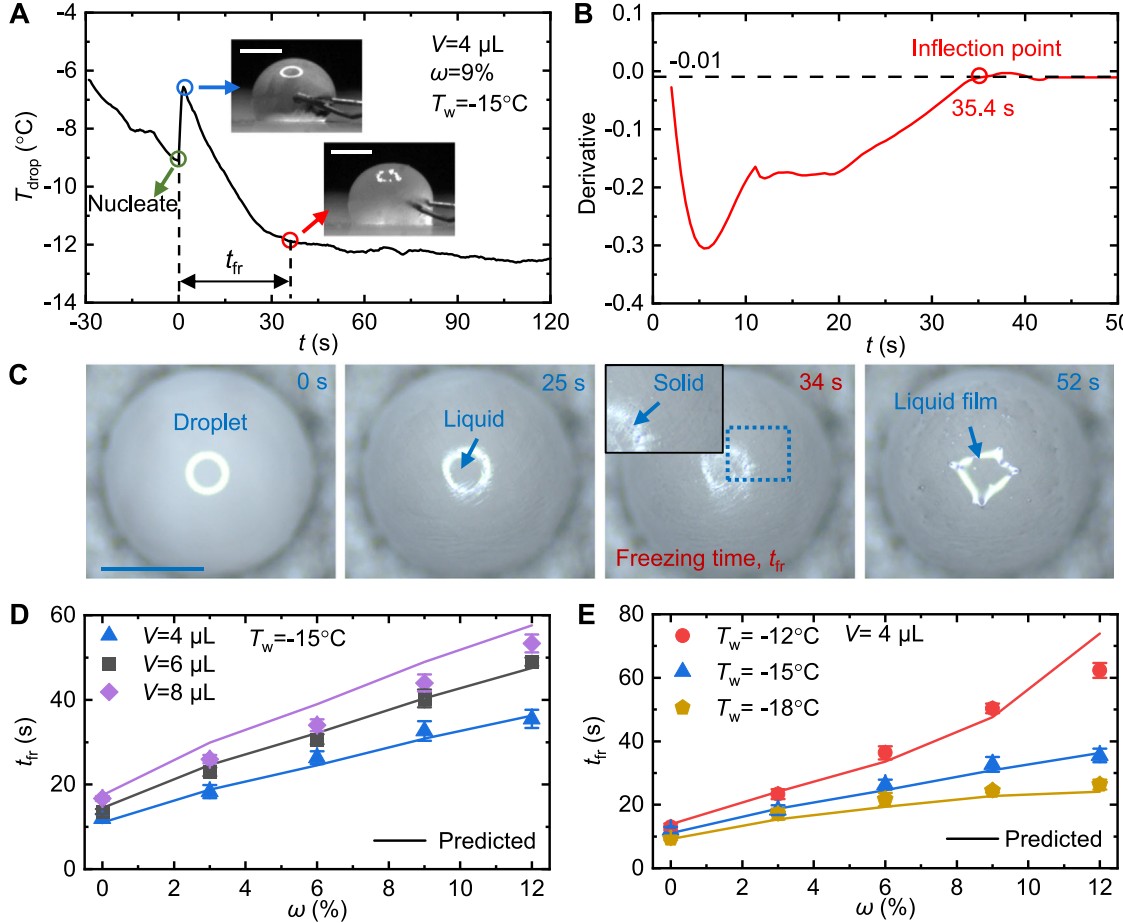

**Fig. 2 | Freezing time of salty droplets. A** Temperature curve of salty droplet icing ($V = 4$ μL, $\omega = 9\%$ and $T_w = -15$ °C). The green circle marks the nucleation point (time zero). The blue circle represents the end of nucleation/recalescence and the nearby inset is a snapshot of the droplet at the same moment. The red circle marks the inflection point, which indicates the end of freezing. The inset beside it shows the droplet snapshot at the same moment. The time duration from the green circle to the red circle is defined as the freezing time ($t_{fr}$). **B** Derivative of the temperature curve smoothed using the Savitzky-Golay method. The temperature curve reaches the inflection point when its derivative approaches zero (the threshold is set to −0.01 here). **C** Top-view images of salty droplet icing ($V = 4$ μL, $\omega = 9\%$, and $T_w = -15$ °C). The moment (34 s) when the liquid film begins to appear coincides with the inflection point in the temperature curve. **D, E** Freezing times $t_{fr}$ under a variety of conditions of $V$, $T_w$, and $\omega$. The points represent experiments, and the lines represent theoretical predictions. Error bars display the standard deviations of parallel measurements. All scale bars represent 1 mm. Source data are provided as a Source Data file.

where $H_f$ is the droplet height or the length of the longest ice crystal aggregate (Figure S9). During salty droplet freezing, $\Delta T$ gradually changes, which differs from the constant value of pure water. Based on Eq. (2), under the premise of examining the change of $\Delta T$, we develop a semi-empirical model to predict the freezing time of salty droplets. See Supplementary Information S8 for additional model details. As observed in Fig. 2D, E, the predicted freezing times agree within 10% of the experimental results.

**Brine film formation and the ice sprouting phenomenon**

To the best of our knowledge, the observed liquid film formation on top of salty droplets after freezing (Fig. 2C) has not been reported in past literature. Since salty droplets cannot completely freeze, and gaps of ice dendrites inside the droplet are filled with concentrated brine, we infer that the liquid film is a brine film that is squeezed out as the ice crystals densify, and the extrusion and formation of the brine film governs the lack of a pointy tip. Based on the slight elevation of the droplet contour line after brine film formation, we estimate the thickness of the brine film to be ~10 μm (Supplementary Information S9, Figure S11). The brine film thickness is affected by multiple factors, i.e., salt concentration and substrate temperature during the early stages of freezing, and air humidity during the later stages of freezing. However, the thickness and its variation cannot be accurately measured or analyzed currently. From the temperature curve, the temperature inside the droplet remains stable after the freezing stage (Fig. 2A), indicating that the concentrated brine inside the droplet is saturated. Using infrared temperature data, we observe that the temperature on the top of the droplet does not change after freezing ends (Figure S3b). Therefore, we conclude that the brine film on top of the droplet is also saturated at its temperature.

Even more interesting is that after brine film formation, ice crystals begin to grow at the bottom of the brine film and finally pierce the film interface (see Supplementary Movie 2). This unique ice-sprouting phenomenon is clearly shown in Fig. 3A, B (marked by the blue arrows). After the ice crystals penetrate the brine film, they continue to grow in the air via desublimation, like the growth of frost crystals on the surface of a frozen pure water droplet[44,45], as shown in Fig. 3A, B (marked by the yellow arrows). Figure 3C shows a schematic of the evolution of a salty droplet after freezing, depicting brine film formation on top of the frozen droplet, ice crystal sprouting at the bottom of the brine film, and frost crystal growth in the air. The observed ice crystal growth from inside the brine film finally causes the frozen salty droplet to morphologically resemble a sprouting tomato[46], which obviously differs from the pointy morphology of a frozen water droplet[15]. Therefore, we name this unique ice crystal growth inside of the brine film the ice sprouting phenomenon.

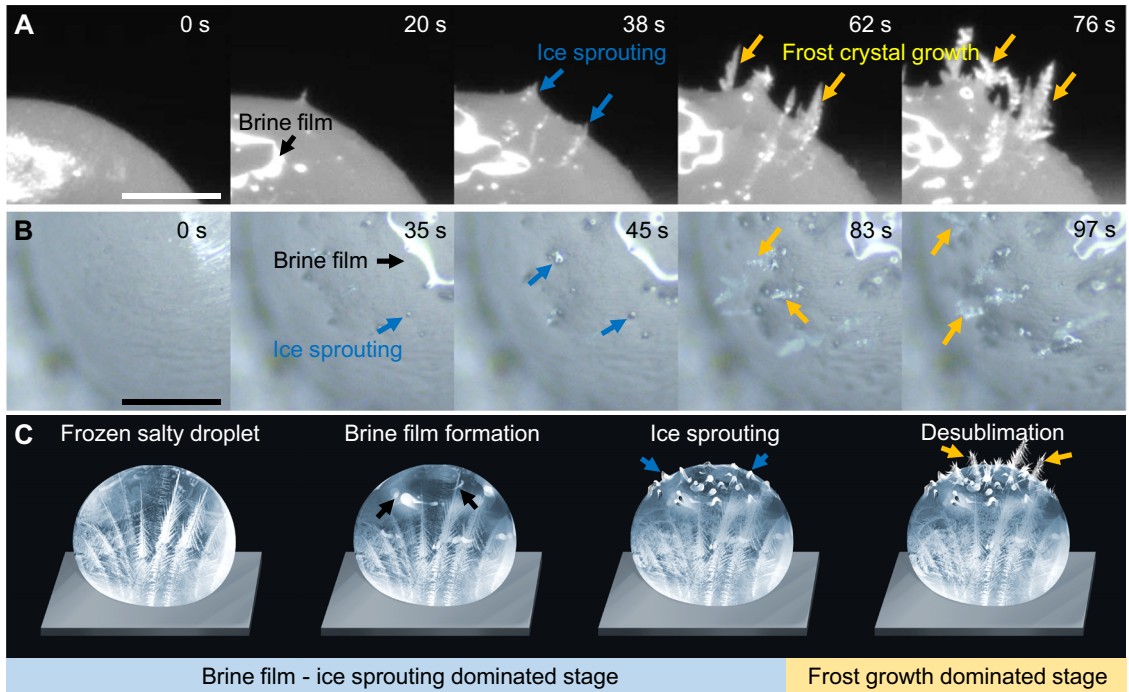

**Fig. 3 | Ice sprouting phenomenon after freezing of a salty droplet. A** Side-view time-lapse images of the salty droplet after its freezing stage ($\omega = 9\%$, $T_w = -15\,°C$, $T_{air} = 25\,°C$, and $RH = 60\%$). The brine film appears on top of the droplet and ice sprouting occurs inside the brine film. The sprouted ice crystals pierce the brine film interface and continue to grow in the air via desublimation. **B** Top-view time-lapse images of the salty droplet after freezing ($\omega = 9\%$, $T_w = -15\,°C$, $T_{air} = 25\,°C$, and $RH = 60\%$). Ice sprouting at the bottom of the brine film and ice crystal growth are clearly shown. **C** Schematic of brine film formation, ice sprouting, and frost growth on a salty droplet. The brine film−ice sprouting dominated stage and frost growth dominated stage are approximately divided and labeled. Brine film formation is indicated by black arrows. Ice sprouting is marked by blue arrows. Frost crystal growth in the air is marked by yellow arrows. The end of freezing is defined as time zero. All scale bars represent 0.5 mm.

Figure 3C can be divided into a brine film−ice sprouting-dominated stage and a frost-growth-dominated stage. During the former, a brine film forms, and ice crystals mainly grow inside the brine film but do not penetrate the brine film extensively. Therefore, the saturated brine film having a stable temperature and concentration can help droplets maintain a stable temperature. Hence why the temperature curves tend to stabilize at the inflection points and are maintained for a period (Fig. 2A, B and Figure S4). However, during the frost growth-dominated stage, many ice crystals pierce the brine film and begin to grow in the air via desublimation. The top of the droplet is covered with frost crystals, which disrupt the continuity of the brine film, leading to its failure to achieve temperature stability. As a result, the temperature curve may begin to fluctuate or decrease.

Although ice sprouting is similar to previously observed frost flower formation on sea ice[47–49], two phenomena are, in fact, different. Ice sprouting refers to the ice crystal growth inside of the brine film. Frozen droplets with brine films are cold, but the surrounding air is at room temperature. In contrast, frost flowers are ice crystals exposed to the air, whose temperature is much lower than the sea ice temperature[49]. The growth processes of frost flowers and ice sprouts also differ. Frost flower formation includes: (i) brine expulsion upwards as sea ice grows; (ii) expelled warm brine forming a mushy or slush layer (i.e., a mixture of ice crystals and brine) after encountering the cold environment; and (iii) mushy layer roughening with some protruding crystals and frost flowers beginning to grow on these protruding crystals[50]. During ice sprouting, the surrounding warm air ensures that the brine will not freeze and will remain clear and optically transparent. Hence, sprouting starts from the ice layer at the bottom of the brine film (Fig. 3B).

### Ice sprouting mechanism
We hypothesize that the ice sprouting mechanism is governed by the maintenance of dynamic stability for the saturated brine film under humid air conditions. As the schematic in Fig. 4A shows, since the brine film temperature is lower than the dew point temperature of the humid air surrounding the droplet, water vapor in the humid air condenses at the brine film interface. The addition of condensed water dilutes the brine film, thereby raising the freezing point of the brine film. As a result, ice crystals precipitate out of the brine film to increase the salt concentration of the brine film and re-saturate the brine film at its temperature. As the condensation continues, ice crystals continue to grow inside the brine film and finally puncture its interface.

To quantitatively prove that the mechanism of the ice sprouting stems from interfacial condensation on the brine film, we design an analogous experiment that aims at measuring the ice precipitation rate inside of a brine film and the condensation rate on the liquid-air interface of the brine film. We first fill a supercooled copper sink with brine that is saturated at the sink temperature. A silver iodide disc is placed into the brine film as a substrate for the precipitation of ice crystals. Since the structure of silver iodide resembles ice, it can trigger ice nucleation[51]. Then, the sink is placed in a humid air environment, and the experiment is started. See Supplementary Information S10 for additional experimental details.

Figure 4B shows ice crystal precipitation on the silver iodide disc. The blue dashed circles mark the location of ice crystals inside the brine film, while the yellow arrows indicate the reflection of the light source, indicating that the ice crystals have penetrated the liquid film (see Supplementary Movie 3). The observed phenomenon represents a good analogy of the ice sprouting, which occurs on top of frozen salty droplets. Several hours into the experiment, we measure the mass rates of precipitated ice and condensed water vapor (Fig. 4C). Under different humidities, we find that the ice precipitation rate and the condensation rate are always consistent, indicating that interfacial condensation governs ice sprout growth. We also calculate the driving

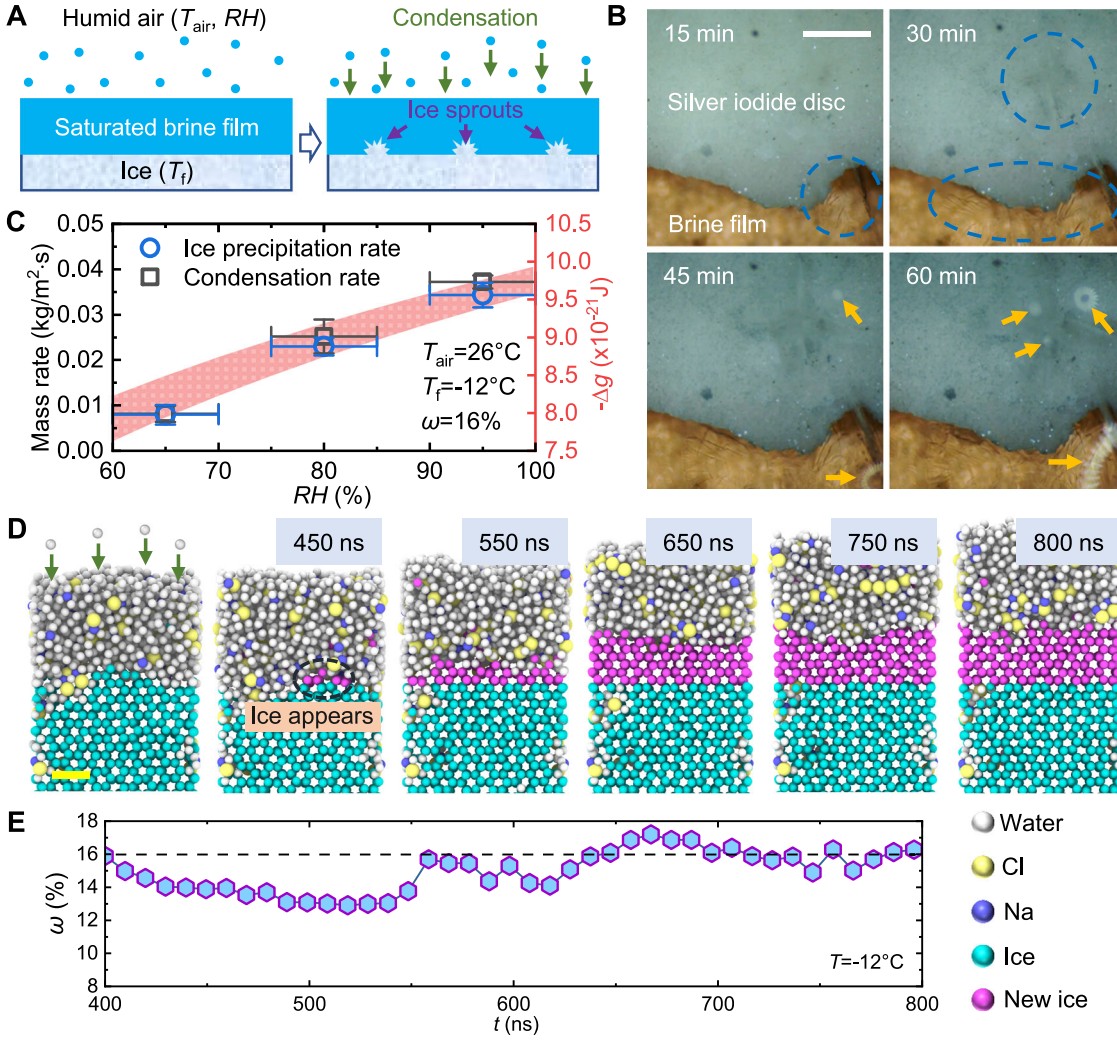

**Fig. 4 | Mechanism of ice sprouting inside the brine film. A** Schematic depicting water vapor in humid air condensing on the liquid-air interface of the saturated brine film, decreasing the salt concentration and making the brine film unsaturated. To re-saturate the brine film, ice sprouts and grows on the ice-liquid interface. **B** Experimental images of ice crystal precipitation on a silver iodide disc. The blue dashed circles mark ice crystals. Yellow arrows indicate ice crystal penetration of the liquid film. Scale bar represents 0.5 mm. **C** Experimentally measured ice precipitation rate and condensation rate as a function of humidity. The shaded area represents the calculated driving potential for interfacial condensation ($-\Delta g$, Eq. (3)). $T_{air}$ and $RH$ represent air temperature and humidity, while $T_f$ represents the copper sink or brine temperature. Error bars display the standard deviations of multiple measurements. **D** Time-lapse images of ice growth MD simulation. Water molecules are added to the salty water to represent condensation after the salty water system thermally stabilizes at $T = 261$ K ($-12$ °C, after 400 ns shown in Fig. 1D). At 450 ns, ice appears, and the newly formed ice (purple color) continues to grow. Scale bar represents 1 nm. **E** Variation of salt concentration for unfrozen liquid versus time during ice growth. The concentration fluctuates slightly around 16% (saturated concentration at $T = -12$ °C), further confirming the mechanism. Source data are provided as a Source Data file.

potential for interfacial condensation under the experimental conditions based on Gibbs free energy analysis:

$$-\Delta g = k_B T_f \ln \frac{p_{v,air}}{p_{v,s}}, \qquad (3)$$

where $-\Delta g$ is the Gibbs free energy decrease denoting the condensation driving potential, $k_B$ is the Boltzmann constant, $p_{v,air}$ is the water vapor partial pressure in humid air, and $p_{v,s}$ is water vapor saturation pressure at brine temperature $T_f$. Substituting the experimental conditions into Eq. (3), we see that the trend of driving potential for interfacial condensation as a function of humidity (pink shaded region in Fig. 4C) agrees well with the experimental data. See Supplementary Information S11 for additional modeling details. In addition, from the perspective of water activity, when the brine film is in equilibrium with the humid air, its water activity can be expressed in terms of the local relative humidity[52]. Hence, when the relative humidity is higher, the

corresponding equilibrium water activity of the brine film is greater. Since the actual activity of the brine film is constant under established conditions, higher humidity causes a greater deviation of the actual activity from equilibrium activity, which would result in a stronger tendency of interfacial condensation to reduce the concentration of the brine film and make its actual activity closer to the equilibrium activity. This qualitative analysis of interfacial condensation is consistent with Gibbs free energy analysis.

We utilize MD simulation to verify the proposed mechanism of the ice sprouting in the brine film from a nanoscale viewpoint. Figure 2D, E, and F show that at a certain temperature, when the concentration of unfrozen brine reaches saturation, icing no longer forms (approximately after 300 ns). Then, we artificially add water molecules into the brine film to model condensation at the liquid-air interface for a period of 400 ns. Within the first 5 ns, 100 water molecules are added, followed by a 35 ns interval. This process is repeated 10 times with a total of 1000 water molecules added. As shown in Fig. 4D

(see Supplementary Movie 4), shortly after the addition of water molecules, ice growth appears in the stable brine film (black dashed circle at 450 ns). According to recent calculations[53], added water molecules rapidly reduce the concentration of the brine film, causing a decrease of the interfacial free energy between ice and solution at a constant system temperature; therefore, new ice begins to grow from the existing frozen surface. During new ice growth, the salt concentration of the brine film fluctuates around 16%, especially in the later stage (Fig. 4E), indicating the dynamic stability of the brine film.

Finally, we compare the differences in mechanisms driving ice sprouting and frost flowering on sea ice. For the ice sprouting inside the brine film, the interfacial condensation breaks the saturation state of the brine film itself, so ice crystals begin to precipitate from the bottom of the brine film to maintain its stability. In other words, the ice sprouting is governed by a condensation-precipitation mechanism. During frost flower formation, evaporation of the mushy layer is the main source of water vapor, allowing for local supersaturation, and subsequent desublimation growth of the frost flower[47–49]. That is, the growth of frost flowers is governed by an evaporation-desublimation mechanism, which differs from the ice-sprouting phenomenon reported here.

## Discussion

In summary, we investigate the characteristics of salty droplet icing on solid surfaces. Differing from water droplets, salty droplet freezing enables the formation of concentrated brine that is squeezed out after the freezing stage. This results in liquid film formation on top of salty droplets that prevent pointy tip formation. By defining the moment of brine film formation as the end of freezing, we propose models to predict the freezing time under various conditions. Surprisingly, we observe ice sprouting inside the brine film, eventually puncturing the film, and resulting in a frozen salty droplet resembling a sprouting tomato in morphology. We qualitatively and quantitatively reveal that the ice sprouting inside the brine film stems from interfacial condensation. The ice sprouting phenomenon occurs under a wide range of air humidity and supercooling conditions unless the air humidity is extremely low to prevent condensation or the surface temperature is lower than the eutectic point temperature (Supplementary Information S12, Figures S14 and S15). The ice sprouting is valid for freezing droplets containing other ions such as $K^+$, $Ca^{2+}$, and $Mg^{2+}$ (Supplementary Information S13, Figure S16). Experiments on hydrophilic surfaces show that all results are not sensitive to the properties of solid surfaces or the shape of droplets (Supplementary Information S14, Figure S17). It should be noted that all the experiments are conducted under conventional natural icing conditions, and some extreme conditions, such as high cooling rates, may change the experimental results by tuning the ion behavior during the salty droplet freezing, which is worthy of further investigation. Overall, our results here elucidate the physics governing salty droplet icing, they enlighten diverse technologies such as freeze desalination and marine anti-icing[39,40]. The mechanism governing the ice sprouting from inside brine films, i.e., the balance between ice precipitation and interfacial condensation, also offers inspiration for the control of crystallization in solutions[54].

## Methods
### Preparation of surfaces and fluids
To enable observation of droplets, hydrophobic surfaces are fabricated and used as experimental surfaces. An acid-salt mixed solution (0.1 mol/L HCl and 0.1 mol/L CuSO$_4$) is used to clean and structure the aluminum substrate for 5–10 min so that micro and nanostructures can be formed. The structured aluminum substrates are immersed into an ethanol solution of 1 wt.% fluoroalkyl silane (1H,1H,2H,2H-Perfluorodecyltriethoxysilane) for 30 min to enable hydrolysis and self-assembly of the silane. After drying in an atmospheric pressure oven at 100 °C for one hour, we obtain the hydrophobic surface. The

surface micro- and nanostructures are characterized by scanning electron microscopy (SEM). The surface wettability is measured by a contact angle meter under both room (25 °C) and supercooled (−15 °C) temperature conditions. See Supplementary Information S1 for more details regarding the fabrication and characterization of the experimental surfaces. In this work, an aqueous solution of sodium chloride (NaCl, CAS#7647-14-5, Aladdin) is mainly prepared as the experimental fluid. The deionized water is used to form the aqueous solution. The mass concentration of NaCl in the salty water ranges from 0 to 16%, which is lower than the eutectic point concentration of NaCl (23.3%). To broaden the universality of our discoveries, aqueous solutions of MgCl$_2$ (CAS#7786-30-3, Aladdin), CaCl$_2$ (CAS#10043-52-4, Aladdin), or KCl (CAS#7447-40-7, Aladdin) are also used. Please refer to the results in Supplementary Information S13. All properties of the experimental fluids are obtained in Lange's Handbook of Chemistry[55].

### Droplet icing experiments
We built a setup for the droplet icing experiments using a semiconductor refrigeration module to provide cooling. In the experiments, we locate the experimental surface on the cold plate and inject droplets onto the surface. The surface temperature can be tuned by changing the input voltage of the semiconductor cooler. Surface temperatures range from −10 to −20 °C in this work, which is below the freezing point of the salty droplets used here. We also set a Hele-Shaw cell, in which a two-dimensional droplet can be produced. Locating the Hele-Shaw cell in the supercooled surface can achieve icing of two-dimensional droplets, which facilitates the observation of ice crystal growth inside the droplet. See Supplementary Information S2 for more experimental details. All icing processes are recorded using a high-speed microphotography module containing a high-speed camera (Photron FASTCAM Mini UX100, Japan) and a microscope (AOSVI 3M150G, China). We analyze the icing phenomena by examining the recorded videos. We also extract quantitative experimental data, such as the freezing time of droplets, from the videos.

### Temperature measurements of icing droplets
The variation of droplet temperature during icing is obtained using thermocouples (Omega TT-T-30, 0.254 mm cable diameter, ±0.5 °C accuracy, USA). The thermocouples (one or two sets) are inserted inside the droplets and measure their internal temperatures. See Supplementary Information S3−S5 for more details on the temperature measurement. We also use an infrared camera (FLIR A615, USA) to measure the surface temperature of icing droplets. During the measurement using an infrared camera, the emissivity is set to a constant value, and the measurement results are calibrated and validated by thermocouples.

### MD simulations
MD simulations are used to provide a physics-based explanation of what is occurring in the experiments from a nanoscale view. The mW-ion model developed by DeMille and Molinero with high accuracy and low computational cost is adopted in the MD simulation[56]. This model can accurately describe the effect of ions on the structure of water, as well as the properties of NaCl solution structure, density, and diffusion coefficient. The force field equations and detailed parameters of the mW-ion model can be found in ref. 57. MD simulations are implemented using the large-scale atomic/molecular massively parallel simulator (LAMMPS). Assuming the initial velocity of the atoms follows the Maxwell-Boltzmann distribution, the Verlet algorithm is used to integrate Newton's equation of motion every 5 fs, and the temperature and pressure are controlled correspondingly by the Nosé-Hoover thermostat and barostat with relaxation times of 0.5 ps and 5 ps, respectively. The atomic structure is visually rendered by OVITO software. The method of coexistence of the NaCl solution and the hexagonal ice, which acts as the seed to trigger the freezing of the NaCl

solution, is applied, and the CHILL+ algorithm is used to distinguish the structure and quantify the number of ice-like and water-like molecules, thereby locating the solid-liquid interface[57]. It should be noted that the nanoscale results obtained by MD simulation cannot be directly compared with the macroscopic experimental results in time and space. See Supplementary Information S6 for more details.

## Analogy experiments of ice crystal precipitation

We designed an analogous experiment to simulate ice sprouting on top of frozen salty droplets. The key module of this setup is a copper sink. We inject brine ($\omega = 16\%$) into the copper sink and immerse a thin silver iodide disc (prepared by pressing silver iodide powder, AgI, CAS#7783-96-2, Aladdin) into the brine film. The brine film is used to simulate the brine on top of a frozen salty droplet, and the silver iodide disc simulates the pre-existing ice at the bottom of the brine film. The structure of AgI resembles ice hence why it is used. The copper sink is cooled to the freezing point of the brine film, and it is placed in a closed chamber, where temperature and humidity are controlled. In the experiment, the copper temperature is set at −12 °C. The chamber temperature is 26 °C, and its humidity is tuned in the range of 65–95%. The chamber humidity is measured by a temperature-humidity sensor (Testo 400, ±0.5 °C accuracy for temperature and ±3% accuracy for humidity, Germany). After the experiment is started, ice crystals gradually grow on the silver iodide disc, and water vapor condensation occurs on the liquid-gas interface of the brine film. We measure the mass of the precipitated ice on the silver iodide disc, and also measure the mass of the condensed water vapor on the brine film surface. After dividing the mass data by the experiment time, we obtain the ice precipitation rate and the condensation rate. See Supplementary Information S10 for more details.

## Data availability

The data that supports the findings of the study are included in the main text and supplementary information files. Source data are provided in this paper.

## Code availability

The codes that support the findings of the study are available from the corresponding authors upon request.

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

## Acknowledgements

The authors gratefully acknowledge funding support from the National Natural Science Foundation of China (no. 52206068, received by F.C., no. 52376061, received by X.W., and no. 52236006, received by Y.F.). We also gratefully acknowledge funding support from the International Institute for Carbon Neutral Energy Research (WPI-I2CNER, received by N.M.), sponsored by the Japanese Ministry of Education, Culture, Sports, Science and Technology.

## Author contributions

F.C., Y.F., and X.Y. conceived the idea for the work during the discussion with X.W. and N.M. Y.F., X.W., and N.M. supervised this work and the overall research direction. F.C. and S.L. fabricated the samples and performed the experiments. C.Z. and Y.L. performed the MD simulations. F.C., Y.F., and X.Y. analyzed the data and developed the theoretical models. F.C., X.Y., and N.M. led the writing the manuscript and subsequent revisions. All authors provided critical feedback and helped shape the manuscript.

## Competing interests

The authors declare no competing interests.
