## [Peer Review File · Nature Communications]

Editorial Note: This manuscript has been previously reviewed at another journal that is not operating a transparent peer review scheme. This document only contains reviewer comments and rebuttal letters for versions considered at *Nature Communications*.Major Comments:

1. “While the pure water icing is fairly well understood, the mechanisms governing the icing of salty water droplets remain shrouded in mystery” – I strongly disagree with this statement. I think the authors should refrain from making statements that can be interpreted by a reader outside the field as if no progress had been made in this area.

There are many aspects of ice growth in salt solutions that have already been explored in field studies, laboratory experiments, simulations, and theory. The topics of interest are typically sea ice formation, the freezing of salty water droplets in the atmosphere, and many more. Following are some papers that can be of interest.

a. The phase evolution of young sea ice [Wettlaufer et. al, GEOPHYSICAL RESEARCH LETTERS (1997) DOI: 10.1029/97GL00877]

b. SOME CRYSTAL GROWTH FEATURES OF SEA ICE [Bennington, K. (1963). Journal of Glaciology, 4(36), 669-688. doi:10.3189/S0022143000028306]

c. Martin, S., Yu, Y., and Drucker, R. (1996), The temperature dependence of frost flower growth on laboratory sea ice and the effect of the flowers on infrared observations of the surface, *J. Geophys. Res.*, 101(C5), 12111–12125, doi:10.1029/96JC00208.

d. Martin, S., Drucker, R., and Fort, M. (1995), A laboratory study of frost flower growth on the surface of young sea ice, *J. Geophys. Res.*, 100(C4), 7027–7036, doi:10.1029/94JC03243.

2. Pertaining to the main comment of Reviewer #1:

“Reviewer # 1:

The manuscript has improved after revision, but my main concerns remain: What is new compared to current knowledge? In addition, my impression is that some conclusions are not fully supported by the experimental observations.”

To me, the main novel finding of this work remains very unclear. To my understanding, what the authors claim to be “ice sprouting” is novel.

“These crystals grow until they pierce the free interface, which we term the ice *sprouting phenomenon*.”

However, a similar phenomenon has been reported in the literature for some time now. It is called the “frost flower” phenomenon.

Briefly,

1. When salty seawater freezes, the highly concentrated salt solution (commonly termed brine or slush since it is highly viscous) is excluded from the ice.
2. The surface of this excluded solution also freezes to form protruding crystals governed by underlying ice or atmospheric vapor deposition.
3. The protruding crystal at the surface is known as “frost flower” and is commonly observed in both sea ice and can be recreated at laboratory conditions.

Some citations below:

1. Roscoe, H. K., Brooks, B., Jackson, A. V., Smith, M. H., Walker, S. J., Obbard, R. W., and Wolff, E. W. (2011), Frost flowers in the laboratory: Growth, characteristics, aerosol, and the underlying sea ice, *J. Geophys. Res.*, 116, D12301, doi:[10.1029/2010JD015144](https://doi.org/10.1029/2010JD015144).

2. Martin, S., Drucker, R., and Fort, M. (1995), A laboratory study of frost flower growth on the surface of young sea ice, *J. Geophys. Res.*, 100(C4), 7027–7036, doi:[10.1029/94JC03243](https://doi.org/10.1029/94JC03243).

3. Martin, S., Yu, Y., and Drucker, R. (1996), The temperature dependence of frost flower growth on laboratory sea ice and the effect of the flowers on infrared observations of the surface, *J. Geophys. Res.*, 101(C5), 12111–12125, doi:[10.1029/96JC00208](https://doi.org/10.1029/96JC00208).

3. Current paper (Abstract) “*We reveal that the ice sprouting is controlled by condensation at the brine film free interface*”. Also, in the subsection titled “*Mechanism of the ice sprouting phenomenon*”

In the manuscript [Martin, S., Drucker, R., and Fort, M. (1995), A laboratory study of frost flower growth on the surface of young sea ice, *J. Geophys. Res.*, 100(C4), 7027–7036, doi:10.1029/94JC03243.] frost flower was grown by supplying water vapor from top.

In other words, modulating water supersaturation can lead to ice growth from the brine, and this has been shown in the frost flower phenomenon before.

**** I think it will be important for the authors to discuss their current results in the context of previous observations of ice growth mechanism in salt water ****

2. Pertaining to the main comment of Reviewer #2:

“Page 3: *Although some preliminary advances of nucleation kinetics and ion repulsion have been reported, the understanding of salty water droplet icing remains limited. What are the unique features of salty droplet icing, how does the freezing morphology of salty droplets differ from that of pure water droplets, and how can one determine and predict the freezing time of salty droplets. The answers to these questions, as well as the revelation of the underlying mechanisms, are knowledge gaps that urgently need to be filled.*”

My Comments:

1. I agree with the author's response that exploring the rate of cooling is beyond the scope of the manuscript. Ultrafast cooling is not typically observed in terrestrial conditions where freezing of salt solutions is relevant.

2. The reply to the question on the inflection point seems clear.

Minor Suggestion:

3. I suggest the authors explain what solid cooling means. (I could not find the definition of that in the text).

3. “Mechanism of the ice sprouting phenomenon”: In this section, the authors use the term “ice sprouting” in MD simulations.

The authors use the term “ice sprouting” for macroscopic ice growth. MD simulations in this work, are performed with a small system that is periodically replicated. Therefore, the length scales of the simulations are orders of magnitude smaller than the macroscopic length scales of experiments. What the authors are modeling is the simple 1D-growth of the prismatic ice plane from a template ice lattice. Therefore, I suggest simply using the term ice growth simulations to explain their results and also in the figure.

4. To me, there is a lack of a clear thermodynamical explanation for interfacial condensation and ice growth.

It is surprising the authors do not refer to some very well-known thermodynamic framework in this work. For example, Koop and coworkers, in their seminar paper (T. Koop, B. P. Luo, A. Tsias and T. Peter, *Nature*, 2000, **406**, 611-614) have shown that that experimental nonequilibrium freezing temperature of aqueous solutions can be predicted entirely from the water activity of the mixture.

I suggest the authors discuss the thermodynamic framework of the interfacial condensation process for the broad audience of Nature Comm.

Grammar Issues:

1. Abstract: “While the pure water icing is **failry** well understood, the mechanisms governing the icing of salty water droplets remain shrouded in mystery.” – Please correct the spelling of **failry**.

Reviewer #2 (Remarks to the Author):

The manuscript is now much clearer in presenting the novelty of the work and also in providing readers with comprehensible wording and descriptions about achievements and weakness of the experiments.

One more thing I would like to point out is that, although the authors claim in Fig. 2A that "the temperature inside the droplet remains stable after the freezing stage" (line 224), the temperature is certainly decreasing as the time elapses from 40 to 120 s. The evidence for the stable temperature might be in Fig. 2B indicating zero derivative after 35 s. However, it displays only up to 50 s; further plots up to 120 s should show negative values. A proper explanation for this decreasing behavior of the temperature is required because this would be important for concluding saturation of brine.

Manuscript ID: NCOMMS-23-43969-T

Title: Interfacial ice sprouting during salty water droplet freezing

Response Letter

In this round of review for Nature Communications, two reviewers have responded to our manuscript and provided very constructive comments. We truly appreciate their efforts and time reading and commenting on our manuscript. We have taken all the reviewers' suggestions, clarified their queries, and carefully revised the manuscript. All changes to the manuscript are highlighted **in blue** and a summary of changes is listed below:

Briefly, in the revised manuscript, we have:

1. Revised some statements in the abstract to avoid misunderstanding amongst the broad audience of the journal.
2. Added discussions about sea ice growth, and clarified differences from the salty droplet freezing reported here.
3. Added discussions about the differences in formation mechanisms between the ice sprouting reported here and the frost flower phenomenon reported in the past.
4. Added explanations for the fluctuation and decrease of the icing droplet temperature following the period of stabilization.
5. Added further thermodynamic explanations of the interfacial condensation and the ice sprouting phenomenon.
6. Revised Figures 3C and 4D, as well as their captions to increase clarity.
7. Deeply analyzed the literature according to the reviewers' suggestions, and supplemented some references appropriately.

This manuscript has undergone three rounds of reviews and revisions. The improvement of the manuscript quality is due to the help of reviewers, so we want to express our sincere gratitude to all the reviewers. Thank you.

Detailed responses to reviewers' comments

Reviewer # 1:

1. “While the pure water icing is fairly well understood, the mechanisms governing the icing of salty water droplets remain shrouded in mystery” – I strongly disagree with this statement. I think the authors should refrain from making statements that can be interpreted by a reader outside the field as if no progress had been made in this area.

There are many aspects of ice growth in salt solutions that have already been explored in field studies, laboratory experiments, simulations, and theory. The topics of interest are typically sea ice formation, the freezing of salty water droplets in the atmosphere, and many more. Following are some papers that can be of interest.

- a. The phase evolution of young sea ice [Wetlaufer et. al, GEOPHYSICAL RESEARCH LETTERS (1997) DOI: 10.1029/97GL00877]
- b. SOME CRYSTAL GROWTH FEATURES OF SEA ICE [Bennington, K. (1963). Journal of Glaciology, 4(36), 669-688. doi:10.3189/S0022143000028306]
- c. Martin, S., Yu, Y., and Drucker, R. (1996), The temperature dependence of frost flower growth on laboratory sea ice and the effect of the flowers on infrared observations of the surface, J. Geophys. Res., 101(C5), 12111–12125, doi:10.1029/96JC00208.
- d. Martin, S., Drucker, R., and Fort, M. (1995), A laboratory study of frost flower growth on the surface of young sea ice, J. Geophys. Res., 100(C4), 7027–7036, doi:10.1029/94JC03243.

Reply: Thank you for your comments. We agree that the term ‘shrouded in mystery’ is not scholarly and have removed it accordingly. We have also carefully read the literature you recommend, which has further deepened our understanding of the basic process of seawater icing, as well as the formation of frost flowers on sea ice. We have added these citations and discussion to the revised manuscript. Based on this literature, we summarize the novelty of our manuscript:

(1) For salty droplet freezing on solid surfaces, we provide a unified definition of the freezing time, from both the droplet temperature variation and the droplet morphology evolution, with a consolidated prediction model. Although sea ice growth research can help us understand the basic process of droplet icing, previous research on sea ice growth cannot directly contribute to the above findings, and these findings do not apply to sea ice growth either.

(2) We observe brine film formation on top of frozen salty droplets and discover a new growth pattern of ice crystals sprouting from the bottom of the brine film, which we term the ‘ice sprouting phenomenon’. This phenomenon causes the droplet to resemble a sprouting tomato in its morphology, and this is a unique freezing morphology compared to the regular pointy frozen water droplet, as shown in **Fig. R1** below.

(3) We reveal the physical mechanism governing the ice sprouting phenomenon to be a balance between interfacial condensation on the brine film and ice precipitation inside the brine film (a condensation-precipitation mechanism), as verified by experiments, free energy analysis, and nanoscale molecular dynamic (MD) simulations.

(4) The ice sprouting phenomenon is different from frost flower growth on sea ice reported in previous literature. The formation conditions, formation process, and formation mechanisms all differ (please

see their detailed differences in our responses to your comment #2). The ice sprouting phenomenon studied here has never been reported in previous literature.

Figure R1. Salty droplet icing demonstrating the ice sprouting phenomenon: comparison among a frozen water droplet, an ice sprouting salty droplet, and a sprouting tomato. Photographs of (a) a frozen water droplet with a pointy tip, (b) side and top views of an ice sprouting salty droplet, and (c) a sprouting tomato demonstrating a similar morphology. The scale bars are 1 mm for all images in (a) and (b).

In addition, according to your comment, we also searched for literature related to atmospheric droplet icing. For atmospheric droplet icing, NaCl aerosols usually serve as ice nucleation particles, which is different from the freezing of salty solution droplets reported in our manuscript.

Overall, we appreciate your kind reminder. To avoid misunderstanding from readers outside the field, we have revised the statement in the abstract as suggested. We have also added discussion about the similarities and differences between salty droplet icing and sea ice growth with proper citations in the revised manuscript.

[Revised contents in the revised Manuscript]

Abstract: While large-scale sea ice growth has been investigated for centuries, the icing features of small salty droplets remain poorly understood.

Page 6: Figure 1G shows images of a frozen salty droplet within a Hele-Shaw cell. The results show that concentrated brines exist in the crevices of ice dendrites, i.e., the frozen droplet is a mixture of ice and concentrated brine. The basic phase-change characteristics (e.g., ice crystal growth and brine expulsion) during salty droplet freezing on a solid surface are similar to that of sea ice growth, except that sea ice grows from the sea surface downwards, leaving brine channels inside the ice⁴¹⁻⁴². Salty droplets heterogeneously nucleate from the solid surface and ice crystals grow upward towards the liquid-vapor interface.

2. Pertaining to the main comment of Reviewer #1: “The manuscript has improved after revision, but my main concerns remain: What is new compared to current knowledge? In addition, my impression is that some conclusions are not fully supported by the experimental observations.”

To me, the main novel finding of this work remains very unclear. To my understanding, what the authors claim to be “ice sprouting” is novel. “These crystals grow until they pierce the free interface, which we term the ice sprouting phenomenon.” However, a similar phenomenon has been reported

in the literature for some time now. It is called the “frost flower” phenomenon.

Briefly,

1. When salty seawater freezes, the highly concentrated salt solution (commonly termed brine or slush since it is highly viscous) is excluded from the ice.
2. The surface of this excluded solution also freezes to form protruding crystals governed by underlying ice or atmospheric vapor deposition.
3. The protruding crystal at the surface is known as “frost flower” and is commonly observed in both sea ice and can be recreated at laboratory conditions.

Some citations below:

1. Roscoe, H. K., Brooks, B., Jackson, A. V., Smith, M. H., Walker, S. J., Obbard, R. W., and Wolff, E. W. (2011), Frost flowers in the laboratory: Growth, characteristics, aerosol, and the underlying sea ice, *J. Geophys. Res.*, 116, D12301, doi:10.1029/2010JD015144.
2. Martin, S., Drucker, R., and Fort, M. (1995), A laboratory study of frost flower growth on the surface of young sea ice, *J. Geophys. Res.*, 100(C4), 7027–7036, doi:10.1029/94JC03243.
3. Martin, S., Yu, Y., and Drucker, R. (1996), The temperature dependence of frost flower growth on laboratory sea ice and the effect of the flowers on infrared observations of the surface, *J. Geophys. Res.*, 101(C5), 12111–12125, doi:10.1029/96JC00208.

Reply: Thank you again for recommending this literature to us. We have carefully read them, as well as other relevant literature (*i.e.*, *Journal of Geophysical Research*, 1994, 99, 16341-16350; *Journal of Geophysical Research*, 2008, 113, D21304), and confirmed that the ice sprouting phenomenon and the underlying mechanism in our work is indeed novel, after comparing this phenomenon with previously reported frost flower phenomenon. The key differences between the two phenomena are listed below.

(1) The conditions for frost flowering and ice sprouting phenomena differ. To initiate frost flower growth, the environmental temperature should be much colder than the sea ice temperature. In contrast, in our experiments, the frozen droplets and expelled brine films are cold, while the surrounding air is at room temperature ($25.0 \pm 3^\circ\text{C}$). The different conditions lead to different formation processes and mechanisms driving the two phenomena, as explained below.

(2) The growth processes of frost flowering and ice sprouting are different. For frost flower formation, as you summarized above, its formation stages include (see **Fig. R2** for the schematic depicting the frost flower growth process on sea ice): (i) brine expulsion upwards as sea ice grows; (ii) mushy or slushy (*i.e.*, a mixture of ice crystals and brine) layer formation by the expelled warm brine after encountering a cold environment; (iii) mushy layer roughening by crystal protrusion leading to frost flower growth on these protruding crystals. In contrast, for the ice sprouting phenomenon: (i) because the surrounding humid air is warm, the expelled brine will not freeze and will remain optically clear and transparent (**Fig. R3**); (ii) because the brine film is colder than the dew point of the humid air, condensation occurs on the upper interface of the brine film and the condensed vapor is the vapor source for the ice sprouting inside the brine film (see **Fig. 4A** for the schematic depicting the ice sprouting mechanism).

Figure R2. Schematic depicting frost flower growth on sea ice. The inserted photo shows the rough mushy brine layer (Roscoe et al., *Journal of Geophysical Research*, 2011, 116, D12301).

Figure R3. Transparent brine film on top of a droplet. The blue arrows show ice sprouting from the bottom of brine film. This figure is reproduced based on Fig. 2C in the manuscript.

(3) The governing mechanisms of frost flower formation and ice sprouting are different. For frost flowering, the evaporation of the mushy layer is a main source of water vapor, allowing for local supersaturation, and desublimation growth of the frost flower. Hence the growth of the frost flower is governed by an evaporation-desublimation mechanism (Fig. R2). However, for ice sprouting inside the brine film, interfacial condensation breaks the saturation state of the brine film itself, so ice crystals begin to precipitate from the bottom of the brine film to maintain its stability. In other words, ice sprouting is governed by a condensation-precipitation mechanism (Fig. 4A), which is confirmed by both our experiments and MD simulations (Fig. 4B-E).

(4) Frost flowers are ice crystals that grow in the air (Fig. R2), while ice sprouting refers to ice crystal precipitation from the liquid (Fig. 4). This is also why the mechanisms of these two phenomena cannot be the same.

With these major differences between our reported phenomena and previously studied frost flowering on sea ice, we conclude that the ice sprouting phenomenon reported in our work is novel. To clarify this important point, and to enable readers to clearly understand the difference between frost flowering and ice sprouting phenomena, we have added new discussions to the revised manuscript.

[Revised contents in the revised Manuscript]

Pages 13-14: Furthermore, one may notice that the ice sprouting phenomenon is similar to previously observed frost flower formation on sea ice⁴⁷⁻⁴⁹. However, these two phenomena are in fact different. Ice sprouting refers to the ice crystal growth inside the brine film. Frozen droplets with brine films are cold but the surrounding air is at room temperature. In contrast, frost flowers are ice crystals that grow in the atmospheric environment, where the atmosphere is colder than the sea ice⁴⁹. The growth processes of frost flowers and ice sprouts also differ. Frost flower formation includes: (i) brine expulsion upwards as sea ice grows; (ii) expelled warm brine forming a mushy or slush layer (i.e., a mixture of ice crystals and brine) after encountering the cold environment; and (iii) mushy layer roughening with some protruding crystals and frost flowers beginning to grow on these protruding crystals⁵⁰. During ice sprouting, the surrounding warm air ensures that the brine will not freeze and will remain clear and optically transparent. Hence, sprouting starts from the ice layer at the bottom of the brine film (Figure 3B).

Page 17: Finally, we compare the differences in mechanisms driving ice sprouting and frost flowering on sea ice. For the ice sprouting inside the brine film, the interfacial condensation breaks the saturation state of the brine film itself, so ice crystals begin to precipitate from the bottom of the brine film to maintain its stability. In other words, the ice sprouting is governed by a condensation-precipitation mechanism. During frost flower formation, evaporation of the mushy layer is the main source of water vapor, allowing for local supersaturation, and subsequent desublimation growth of the frost flower⁴⁷⁻⁴⁹. That is, the growth of frost flower is governed by an evaporation-desublimation mechanism, which differs from the ice sprouting phenomenon reported here.

3. Current paper (Abstract) “We reveal that the ice sprouting is controlled by condensation at the brine film free interface”. Also, in the subsection titled “Mechanism of the ice sprouting phenomenon”.

In the manuscript [Martin, S., Drucker, R., and Fort, M. (1995), A laboratory study of frost flower growth on the surface of young sea ice, *J. Geophys. Res.*, 100(C4), 7027–7036, doi:10.1029/94JC03243.] frost flower was grown by supplying water vapor from top.

In other words, modulating water supersaturation can lead to ice growth from the brine, and this has been shown in the frost flower phenomenon before.

**** I think it will be important for the authors to discuss their current results in the context of previous observations of ice growth mechanism in salt water ****

Reply: Thank you for your comment. For frost flower growth, a local supersaturation above the mushy brine layer is necessary. However, although the additional vaporizer supplies a part of water vapor, this part of water vapor is not the main source of water vapor that ensures local supersaturation. As mentioned above in the reply to comment #2, the growth of frost flowers is governed by an evaporation-desublimation mechanism (**Fig. R2**). The evaporation of the mushy brine layer provides the main source of water vapor to reach local supersaturation.

Martin *et al.* supported this conclusion by saying “A surface water budget for the flowers and slush layer shows that most of the water in the flowers and slush layer comes from the ice interior, not from the vaporizer” (*Journal of Geophysical Research*, 1995, 100, 7027–7036), and “The evaporation of vapor from this liquid into the atmospheric boundary layer provided the supersaturated region

adjacent to the ice surface” (*Journal of Geophysical Research*, 1996, 101, 12111–12125). The paper by Perovich and Richter-Menge also supported this by stating “This suggests that while the water vapor from open water may contribute to creating the conditions necessary for frost flowers, it alone is not always sufficient”, and “We believe that the brine skim further contributed to frost flower development by providing a source of surplus water vapor necessary for their growth” (*Journal of Geophysical Research*, 1994, 99, 16341-16350).

The ice sprouting inside the brine film reported in our work is governed by a condensation-precipitation mechanism (**Fig. 4A**), and the water vapor in the surrounding humid air at room temperature is the only source for ice sprouting. Therefore, in terms of mechanism, there is a fundamental difference between the phenomena of ice sprouting and frost flowering. In the revised manuscript, we have provided detailed discussions about the formation mechanisms of these two phenomena, as well as the differences between them.

[Revised contents in the revised Manuscript]

Page 17: *Finally, we compare the differences in mechanisms driving ice sprouting and frost flowering on sea ice. For the ice sprouting inside the brine film, the interfacial condensation breaks the saturation state of the brine film itself, so ice crystals begin to precipitate from the bottom of the brine film to maintain its stability. In other words, the ice sprouting is governed by a condensation-precipitation mechanism. During frost flower formation, evaporation of the mushy layer is the main source of water vapor, allowing for local supersaturation, and subsequent desublimation growth of the frost flower⁴⁷⁻⁴⁹. That is, the growth of frost flower is governed by an evaporation-desublimation mechanism, which differs from the ice sprouting phenomenon reported here.*

4. Pertaining to the main comment of Reviewer #2: “Page 3: Although some preliminary advances of nucleation kinetics and ion repulsion have been reported, the understanding of salty water droplet icing remains limited. What are the unique features of salty droplet icing, how does the freezing morphology of salty droplets differ from that of pure water droplets, and how can one determine and predict the freezing time of salty droplets. The answers to these questions, as well as the revelation of the underlying mechanisms, are knowledge gaps that urgently need to be filled.”

My Comments:

1. I agree with the author's response that exploring the rate of cooling is beyond the scope of the manuscript. Ultrafast cooling is not typically observed in terrestrial conditions where freezing of salt solutions is relevant.
2. The reply to the question on the inflection point seems clear.

Minor Suggestion:

3. I suggest the authors explain what solid cooling means. (I could not find the definition of that in the text).

Reply: First, we thank the reviewer for agreeing with our viewpoint. Second, we agree with the reviewer’s suggestion and explain the solid cooling stage for pure water droplet icing here.

In our response to comment #3 of the previous Reviewer #2, we mentioned a solid cooling stage

closely (in time) after the freezing stage of a pure water droplet. We stated that it is also an important difference between salty droplet icing and pure water droplet icing, because there is no instant solid cooling stage for salty droplet icing. As shown in **Fig. 1A** and related contents, for a pure water droplet located on a cold substrate, its icing begins with the nucleation/recalcescence stage, and then the freezing stage. At the end of the freezing stage, a pointy tip forms on the top of droplet (**Fig. 1A**). For a pure water droplet, once the pointy tip is formed, the droplet becomes a complete solid ice droplet. Because the substrate is colder than the ice droplet, the temperature of the solid ice droplet continues to decrease (**Fig. R4**), and this temperature decrease stage after the freezing stage is called the solid cooling stage, as demarked by the green background in **Fig. R4**. To clarify this important point, we have explained the solid cooling stage for pure water droplet icing in the revised Supplementary Information.

Figure R4. Temperature curves and stages during icing of a pure water droplet ($V = 4 \mu\text{L}$ and $T_w = -15^\circ\text{C}$). After the freezing stage (i.e., the inflection point), it is the solid cooling stage (i.e., the slope of the temperature curve is not zero), during which that ice droplet temperature decreases.

[Revised contents in the revised Supplementary Information]

Page S-6: For salty droplet icing, the inflection point means the slope of the temperature curve becomes zero; while for pure water droplet icing, the inflection point indicates a change in the slope of the temperature curve (Figure S3a). This is because a pure water droplet undergoes a solid cooling stage after the freezing stage ends. The term solid cooling refers to the temperature of the solid ice droplet continuing to decrease because of the colder surface (see red line in Figure S3a). There is no instant solid cooling stage for salty droplet icing, an important difference between salty droplet icing and pure water droplet icing.

5. “Mechanism of the ice sprouting phenomenon”: In this section, the authors use the term “ice sprouting” in MD simulations.

The authors use the term “ice sprouting” for macroscopic ice growth. MD simulations in this work, are performed with a small system that is periodically replicated. Therefore, the length scales of the simulations are orders of magnitude smaller than the macroscopic length scales of experiments. What the authors are modeling is the simple 1D-growth of the prismatic ice plane from a template ice lattice. Therefore, I suggest simply using the term ice growth simulations to explain their results and also in the figure.

Reply: Thank you for your suggestion. We agree and have revised the manuscript to use the term “ice growth simulations” to explain the MD simulation results in related texts and figure captions.

6. To me, there is a lack of a clear thermodynamical explanation for interfacial condensation and ice growth. It is surprising the authors do not refer to some very well-known thermodynamic framework in this work. For example, Koop and coworkers, in their seminar paper (T. Koop, B. P. Luo, A. Tsias and T. Peter, *Nature*, 2000, 406, 611-614) have shown that that experimental nonequilibrium freezing temperature of aqueous solutions can be predicted entirely from the water activity of the mixture.

I suggest the authors discuss the thermodynamic framework of the interfacial condensation process for the broad audience of *Nature Comm*.

Reply: Thank you for recommending this important comment and paper to us. In this paper, Koop and coworkers showed that the homogeneous nucleation of ice from supercooled aqueous solutions depends only on the water activity of the solution, and the presence of solutes and the application of pressure have a similar effect on the homogeneous ice nucleation. This paper developed a unified thermodynamic framework for homogeneous ice nucleation regarding the effects of solutes and pressure, which has significant impacts on subsequent research. Although the homogeneous nucleation that this *Nature* paper focused on is not applicable to our work (i.e., the salty droplet icing on solid surfaces in our work is initiated by heterogeneous nucleation), it provides us with great inspiration to enrich our discussions in the manuscript.

Turning back to the ice sprouting inside the brine film, our experiments and MD simulation have demonstrated that this mechanism is governed by the maintenance of dynamic stability for the saturated brine film under humid air conditions. According to the *Nature* paper, when the brine film is in equilibrium with humid air, its water activity may be equivalently expressed in terms of local relative humidity. That means, when the relative humidity is higher, the corresponding equilibrium water activity of the brine film is greater. As we all known, a thermodynamic system always tends towards an equilibrium state. Since the actual activity of the brine film is constant under established conditions, higher humidity causes a greater deviation of the actual activity from equilibrium activity, which would result in stronger tendency of interfacial condensation to reduce the concentration of brine film and make its actual activity close to the equilibrium activity. This qualitative analysis of the interfacial condensation tendency is consistent with the Gibbs free energy analysis.

We have added the above discussion to the revised manuscript. To further strengthen the analysis in the manuscript, we have also properly added other thermodynamic bases (e.g., the interfacial free energy between ice and solution) to help elucidate the ice sprouting phenomenon.

[Revised contents in the revised Manuscript]

Page 16: *In addition, from the perspective of water activity, when the brine film is in equilibrium with the humid air, its water activity can be expressed in terms of the local relative humidity⁵². Hence, when the relative humidity is higher, the corresponding equilibrium water activity of the brine film is greater. Since the actual activity of the brine film is constant under established conditions, higher humidity causes a greater deviation of the actual activity from equilibrium activity, which would result in a stronger tendency of interfacial condensation to reduce the concentration of the brine film and*

make its actual activity closer to the equilibrium activity. This qualitative analysis of interfacial condensation is consistent with Gibbs free energy analysis.

Page 17: *As shown in Figure 4D (see Supplementary Movie S4), shortly after the addition of water molecules, ice growth appears in the stable brine film (black dashed circle at 450 ns). According to recent calculations⁵³, added water molecules rapidly reduce the concentration of the brine film, causing a decrease of the interfacial free energy between ice and solution at a constant system temperature; therefore, new ice begins to grow from the existing frozen surface. During new ice growth, the salt concentration of the brine film fluctuates around 16%, especially in the later stage (Figure 4E), indicating the dynamic stability of the brine film.*

7. Grammar Issues:

Abstract: “While the pure water icing is failry well understood, the mechanisms governing the icing of salty water droplets remain shrouded in mystery.” – Please correct the spelling of failry.

Reply: Thank you for your comment. We have revised this sentence in the abstract according to your first comment. We have also carefully checked the spelling and grammar issues in the manuscript.

[Revised contents in the revised Manuscript]

Abstract: *While large-scale sea ice growth has been investigated for centuries, the icing features of small salty droplets remain poorly understood.*

Reviewer # 2:

The manuscript is now much clearer in presenting the novelty of the work and also in providing readers with comprehensible wording and descriptions about achievements and weakness of the experiments.

One more thing I would like to point out is that, although the authors claim in Fig. 2A that "the temperature inside the droplet remains stable after the freezing stage" (line 224), the temperature is certainly decreasing as the time elapses from 40 to 120 s. The evidence for the stable temperature might be in Fig. 2B indicating zero derivative after 35 s. However, it displays only up to 50 s; further plots up to 120 s should show negative values. A proper explanation for this decreasing behavior of the temperature is required because this would be important for concluding saturation of brine.

Reply: First, we want to thank you for your approval of our revision. Second, we are really grateful for your efforts in reviewing our manuscript three times. Your comments have encouraged us to develop a deeper understanding on the discovered phenomena. We thank you for that.

As suggested by your comment, we have examined the temperature curves for longer durations up to 120 s. As shown in **Fig. R5a**, we agree with you that after the inflection points (i.e., the freezing end of salty droplets, marked as red circles), the stable temperature ranges on the temperature curves (marked with green backgrounds) are not long-lasting, and then the temperatures may fluctuate or begin to slightly decrease, as marked with blue backgrounds.

To explain this phenomenon, in **Fig. R5b** (reproduced based on **Fig. 3C**), the brine film - ice sprouting dominated stage and the frost growth dominated stage are divided. During the first stage, brine film forms and ice begins to sprout inside the brine film. It should be noted that, at this stage, ice sprouts mainly grow inside the brine film and do not penetrate the brine film extensively. Therefore, the saturated brine film, having a stable temperature and concentration, can help droplets maintain a stable temperature.

However, during the frost growth dominated stage, many ice crystals pierce the brine film and begin to grow in the air via desublimation. At this time, the top of the droplet is covered with frost crystals, which disrupt the continuity of the brine film, leading to its failure to achieve temperature control. As a result, the temperature inside the droplet begins to fluctuate or decrease. Dense frost crystals can also enhance the heat exchange between droplets and cold surfaces, which may also be the reason for the decrease in droplet temperature.

We are currently unable to quantitatively distinguish these two stages. We can only provide qualitative explanations for the later temperature fluctuation or decrease. To address this important point, we have pointed out this temperature variation and supplemented the above explanation in the revised manuscript.

Figure R5. (a) Reproduced figure based on **Fig. 2A** and **Fig. S4**. The salty droplet icing conditions are $V = 4 \mu\text{L}$, $\omega = 9\%$, and $T_w = -15^\circ\text{C}$. (b) Reproduced schematic based on **Fig. 3C**. In this schematic, brine film - ice sprouting dominated stage and frost growth dominated stage are divided.

[Revised contents in the revised Manuscript]

Page 13: *Figure 3C can be divided into a brine film - ice sprouting dominated stage and a frost growth dominated stage. During the former, a brine film forms, and ice crystals mainly grow inside the brine film but do not penetrate the brine film extensively. Therefore, the saturated brine film having a stable temperature and concentration can help droplets maintain a stable temperature. Hence why the temperature curves tend to stabilize at the inflection points and are maintained for a period (Figure 2A-B and Figure S4). However, during the frost growth dominated stage, many ice crystals pierce the brine film and begin to grow in the air via desublimation. The top of the droplet is covered with frost crystals, which disrupt the continuity of the brine film, leading to its failure to achieve temperature stability. As a result, the temperature curve may begin to fluctuate or decrease.*

REVIEWERS' COMMENTS

Reviewer #1 (Remarks to the Author):

The authors put satisfactory efforts to describe the difference between sea ice and frost flower growth, which is highly appreciated. I am satisfied with the revised manuscript and all my points are carefully addressed (further review from my part is not needed)

Minor comments for the revised manuscript:

1. Abstract: "While large-scale sea ice growth has been

32 investigated for centuries, the icing features of small salty droplets remain poorly understood."

Comment: I think it should be for decades not centuries. (Most of the modern geophysics research started post 1970s)

2. Page 13: " In contrast, frost flowers are ice crystals that grow in the atmospheric environment, where the atmosphere is

colder than the sea ice⁴⁹ "

Comment: please clarify what is atmospheric environment in this case (possibly revise the sentence to mention the specific conditions).

3. Page 13: "During ice sprouting, the surrounding

warm air ensures that the brine will not freeze"

Comment: This is because of brine have lower freezing point ? (colligative property) .. what is the temperature differential between freezing point of the freezing excluded brine and the air temperature?

Reviewer #2 (Remarks to the Author):

I think that the manuscript is now revised well as for my last comment after the authors defined ice sprouting stage and frost growth stage, even though the definition is qualitative. In this respect, comments 2 and 3 of Reviewer #1 were very much helpful also for me to gain deeper understanding. I would recommend publishing this article if Reviewer #1 does as well.

Manuscript ID: NCOMMS-23-43969A

Title: Interfacial ice sprouting during salty water droplet freezing

Response Letter

Detailed responses to reviewers' comments

Reviewer # 1:

The authors put satisfactory efforts to describe the difference between sea ice and frost flower growth, which is highly appreciated. I am satisfied with the revised manuscript and all my points are carefully addressed (further review from my part is not needed).

Reply: Thank you very much for your suggestions, which deepen our understanding of seawater droplet freezing and have helped us improve our manuscript.

Minor comments for the revised manuscript:

1. Abstract: "While large-scale sea ice growth has been investigated for centuries, the icing features of small salty droplets remain poorly understood."

Comment: I think it should be for decades not centuries. (Most of the modern geophysics research started post 1970s)

Reply: Thank you for your suggestion. We agree with you and have replaced "centuries" with "decades" in the abstract.

2. Page 13: "In contrast, frost flowers are ice crystals that grow in the atmospheric environment, where the atmosphere is colder than the sea ice"

Comment: please clarify what is atmospheric environment in this case (possibly revise the sentence to mention the specific conditions).

Reply: Thank you for your suggestion. Frost flowers grow in the atmospheric environment means they grow in the air. While the ice sprouting phenomenon in this work refers to the sprouting of ice crystals in liquid (brine). To clarify this, we have revised the sentence to "In contrast, frost flowers are ice crystals exposed to the air, whose temperature is much lower than the sea ice temperature".

3. Page 13: "During ice sprouting, the surrounding warm air ensures that the brine will not freeze".

Comment: This is because of brine have lower freezing point? (Colligative property). What is the temperature differential between freezing point of the freezing excluded brine and the air temperature?

Reply: Thank you for your comment. In our experiments, the surrounding air temperature is $25.0 \pm 3^\circ\text{C}$, which is much higher than the freezing point of the brine. In other words, the temperature difference between the air temperature and the freezing point of the excluded brine is so large that the brine cannot freeze.

Reviewer # 2:

I think that the manuscript is now revised well as for my last comment after the authors defined ice sprouting stage and frost growth stage, even though the definition is qualitative. In this respect, comments 2 and 3 of Reviewer #1 were very much helpful also for me to gain deeper understanding. I would recommend publishing this article if Reviewer #1 does as well.

Reply: Thank you for your approval of our work. Your efforts in reviewing our manuscript have helped us to greatly improve the quality of the manuscript, which is highly appreciated.